# Assessing Geotourism Resources on a Local Level: A Case Study from Southern Moravia (Czech Republic)

**Lucie Kubalíková**

Department of Geology and Pedology, Faculty of Forestry and Wood Technology, Mendel University in Brno, Zemědělská 3, 61300 Brno, Czech Republic; Lucie.Kubalikova@ugn.cas.cz

**Abstract:** In the last decades, the geotourism has shown a considerable growth all over the world and it is appreciated and accepted as a useful tool for promoting natural and cultural heritage and for fostering local and regional economic development, especially within rural areas. Geotourism focus especially on the geological and geomorphological aspects of the landscape; however, according to the current holistic approach, it also builds on the close relations between geodiversity and other assets of the territory, such as biodiversity, archaeological and cultural values, gastronomy or architecture. Currently, geotourism activities are promoted mainly within geoparks, but other regions also possess an important geotourism potential. A complex assessment of the geotourism resources of a particular area is crucial for geotourism-development. The paper presents two case studies from Southern Moravia (Czech Republic) where the assessment of geotourism's potential was made by using the geomorphosite concept and extended SWOT analysis. Results show that these areas (situated outside the geoparks or large-scale protected areas and not far from a big city) have considerable potential for geotourism development, and geodiversity can be considered an important resource for local and regional development. Based on this, conclusions about the possibilities of geotourism development outside the geoparks are outlined.

**Keywords:** geodiversity; SWOT analysis; rural regions; geomorphosites

## 1. Introduction

In the last few decades, the geotourism has shown considerable growth all over the world [1–4]. Originally, and in a strict sense, the geotourism was defined in the 1990s as "the provision of interpretive and service facilities to enable tourists to acquire knowledge and understanding of the geology and geomorphology of a site (including its contribution to the development of the Earth sciences) beyond the level of mere aesthetic appreciation" [5]. A similar approach was presented by Słomka and Kicińska-Świderska [6], Joyce [7], and Dowling and Newsome [8].

In the broader sense, the geotourism is understood as a form of nature-tourism that focuses on landscape and geology, but also on the biotic and cultural features that are linked to the abiotic nature [9]. It is a so-called ABC (abiotic—biotic—culture) approach. This approach is also reflected in Arouca declaration [10] where geotourism is defined as "tourism which sustains and enhances the identity of a territory, taking into consideration its geology, environment, culture, aesthetics, heritage and the well-being of its residents." The economic and environmental aspects of geotourism are emphasized as well: Dowling [9] defines geotourism as "sustainable tourism with a primary focus on experiencing the Earth's geologic features in a way that fosters environmental and cultural understanding, appreciation and conservation, and is locally beneficial. Geotourism product protects, communicates and promotes geoheritage, helps build communities and works with a wide range of different people." Martini et al. [11] present a more literal and comprehensible definition: "Geotourism allows tourists to know the local geology but also to better understand that this geology is closely

related to all the other assets of the territory, such as biodiversity, archaeological and cultural values, gastronomy, etc." The growing interest in geotourism and the interdisciplinary approach adopted for geotourism studies is reflected in the increasing number of papers and the wide scope of particular topics; e.g., geocultural heritage, geotourism's role in regional development or geotourist perceptions [4].

Currently, this holistic approach is widely respected [2], but geological, geomorphological, pedological and hydrological aspects (the components of geodiversity as defined by Gray [12]) stay in the center of attention and represent the basic resource for geotourist activities. Nevertheless, it has to be remembered that setting the links between geodiversity, biodiversity, culture, and history can help to appreciate the geodiversity as a full-value resource for tourist activities, and thus, as an important resource for local and regional development. This approach is widely applied, especially within geoparks, which are defined as areas with particular geological heritage and a sustainable territorial development strategy [13,14]. This is also the case of the Central European countries, including the Czech Republic: Geotourist activities are developed in geoparks [15,16] and in some cases, in large-scale protected areas such as National Parks or Protected Landscape Areas [17]. However, outside the geoparks, the geodiversity represents an important resource for geotourism development too (see case studies in [2,8,18]).

Two study areas in the South Moravian Region (shortly Southern Moravia) in the southeastern part of the Czech Republic were assessed by using the selected criteria within the geomorphosite concept [19,20] and extended SWOT analysis [21]. These areas are not a part of any geoparks or large-scale protected areas in the sense of the Law 114/1992 Coll [22]. The areas of interest (Deblínská vrchovina Highlands and Sýkořská hornatina Mountains) were already a subject of scientific research, including the description of the potential sites of geoconservation and geotourist interest [23–27]; however, a complex assessment of geotourism resources was not elaborated—only the pilot assessment of geotourism resources of these areas and several sites was a subject of conference papers [23,24,28]. In these terms, the article brings a more complex view on the geotourism resources and their potential in these areas.

## 2. Materials and Methods

### 2.1. Assessment of the Geotourism Potential

To recognize the potential of an area for geotourism, it is necessary to undertake the detailed literature and map review and detailed fieldwork which takes into account both abiotic resources (geodiversity) and other types of resources that are related to geodiversity (biotic, cultural aspects) [18]. Numerous methods for assessing the geotourism potential have been developed (for an overview the works of [20,29–34] are relevant), but they have been limited to an assessment of particular geological or geomorphological sites. Larger areas were assessed by the methods using the GIS-based analysis, e.g., [35–39], but those procedures were usually focused on geoheritage management or implications for geoconservation and did not include the cultural or economical aspects that are essential for geotourism development.

As the geomorphosites are defined as landforms that have acquired a scientific, cultural/historical, aesthetic and/or social/economic value due to human perception or exploitation, and these landforms can be represented both by single geomorphological objects and wider landscapes [19,40], it is supposed that the criteria used within this concept for the assessment of single geomorphological objects can be applied for the qualitative assessment of larger areas ("wider landscapes") as well.

Within the geomorphosite assessment, the assessment criteria are usually divided into several groups (e.g., [19,20,41–44]): Scientific value, added value, economic value, and conservation value. For the qualitative and semi-quantitative assessment of the areas and specific sites of geotourist interest, the method proposed by Reynard et al. [20] is used (Table 1). This method was already applied in several cases; e.g., [45]. It includes all the groups of the values which correspond with a holistic approach to geotourism and respect five pillars of geotourism defined by [18]. The criteria for the site

assessment are applied without changes and numerical scoring is used. In the case of the assessment of "wider landscapes," some criteria were excluded, adapted and assessed qualitatively. The qualitative assessment is based on the detailed literature review, fieldwork and partly on the discussions with local people and it takes into account the assessment of particular sites.

**Table 1.** Criteria used for the qualitative and semi-quantitative assessment of the geotourism potential of the study areas and selected sites of geotourism interest (based on [20,42,45]).

| | Criterion | Brief Explication |
|---|---|---|
| scientific value | integrity | current status of the site or area, degree of degradation of Earth-science features; assessed on the scale from 0 (null) to 1 (excellent); for the areas, the overall landscape quality is assessed |
| | representativeness | the site's or area's exemplarity; assessed on the scale from 0 (null) to 1 (excellent) |
| | rareness | the existence of features that are unique on the national level; assessed on the scale from 0 (null) to 1 (excellent) |
| | paleogeographical interest | importance of the site for the Earth or climate history; the sites assessed on the scale from 0 (null) to 1 (excellent); this criterion was not applied for the assessment of the areas |
| | synthesis | the average of the values (applicable for site assessment) |
| added value | ecological | specific or rare species, important ecosystems; assessed on the scale from 0 (no ecological value) to 1 (high ecological value) |
| | aesthetical | viewpoints, contrasts, space structuration; assessed on the scale from 0 (no aesthetical value) to 1 (high aesthetical value) |
| | cultural | archaeological, historical, artistic aspects of the area related to geodiversity, anthropogenic landforms; assessed on the scale from 0 (no cultural value) to 1 (high cultural value) |
| | synthesis | the average of the values (applicable for site assessment) |
| use characteristics | protection status | legal protection and conservation of the Earth-science features, the sites assessed on the scale from 0 (no protection) to 1 (Earth-science feature as a subject of protection) |
| | threats | risks and hazards: threats to geodiversity—both anthropogenic and natural, assessed on the scale from 0 (existing threats) to 1 (no considerable threats) |
| | accessibility | both by public and individual transport, location of the transport facilities in the proximity (in the case of sites); assessed on the scale from 0 (site with a limited accessibility) to 1 (site with a very good accessibility); for the areas, the "permeability of the landscape" is taken into account |
| | security | safety and limitations on specific sites, assessed on the scale from 0 (problems with safety) to 1 (no considerable limitations); this criterion was not applied to the area assessment |
| | site context | applicable only on the site assessment |
| | tourist infrastructure | catering, accommodation, shelters, tourist paths leading to the sites, proximity of these features to the specific sites; assessed on the scale from 0 (missing infrastructure) to 1 (present and diverse infrastructure) |
| | interpretive facilities | existing interpretive facilities, promotion of the sites/area, supporting products, the common knowledge of the area, assessed on the scale from 0 (missing interpretive facilities) to 1 (present and diverse interpretive facilities) |
| | educational interest | the potential for interpretation, comprehensibility for the lay public; assessed on the scale from 0 (low potential for interpretation) to 1 (high potential for interpretation) |
| | synthesis | the average of the values (applicable for site assessment) |

The assessment is accompanied by extended SWOT analysis (Table 2) which is widely used as a common tool for local development strategies. Basic SWOT analysis has been already employed for the assessment of geotourist resources, e.g., [23,44,46,47], but extended SWOT analysis (or so-called "TOWS matrix") offers a more complex view on the geotourist resources as it provides important information about the applicability and feasibility of geotourism-development [21].

**Table 2.** Extended SWOT analysis ("TOWS matrix").

|  | Strengths | Weaknesses |
|---|---|---|
| Opportunities | Strengths—Opportunities (S-O) strategy (maxi-maxi): use strengths to take advantage of opportunities | Weaknesses—Opportunities (W-O) strategy (mini-maxi): overcome weaknesses by taking advantages of opportunities |
| Threats | Strengths—Threats (S-T) strategy (maxi-mini): use strengths to avoid the threats | Weaknesses—Threats (W-T) strategy (mini-mini): minimize weaknesses and avoid threats |

Coming out of this complex assessment, the possibilities of geotourism-development are presented.

### 2.2. Study Areas

The geotourist potential was analyzed in the two areas of interest: The Sýkořská hornatina Mountains (the southern part) and Deblínská vrchovina Highland (Figure 1) which both belong to the geological unit of Svratka Dome ([27,48], Figure 2). Between these areas, Tišnovská kotlina and Šerkovická kotlina basins are situated and form a natural connection between the two areas. These areas are not a part of any geoparks or large-scale protected area (Protected Landscape Area or National Park according to the Czech legislative, [22]); they are of rural character and they already partly serve as a recreation base for people from the Moravian metropolis Brno and nearby towns.

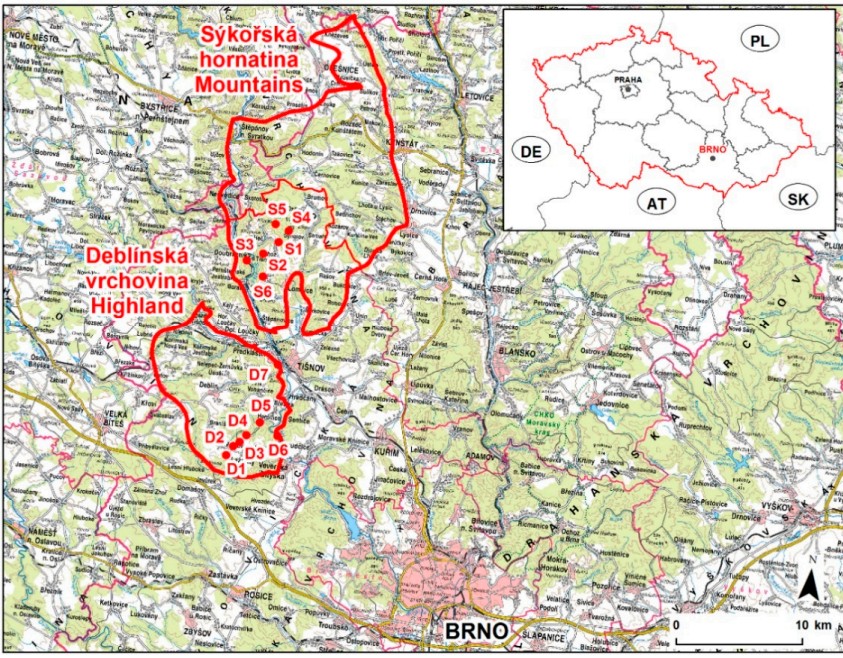

**Figure 1.** Position of the study areas within the southeastern part of the Czech Republic and selected sites of geotourist-interest: S1—Dobrá studně, S2—Hrušín, S3—pod Sokolí skálou, S4—Synalovské kopaniny, S5—Míchovec, S6—Veselský chlum, D1—Skalky, D2—kaolin pit, D3—abandoned limestone quarry, D4—karst spring, D5—Marškovský and Pejškovský potok streams, D6—Svratka valley, and D7—Vokoun's viewpoint.

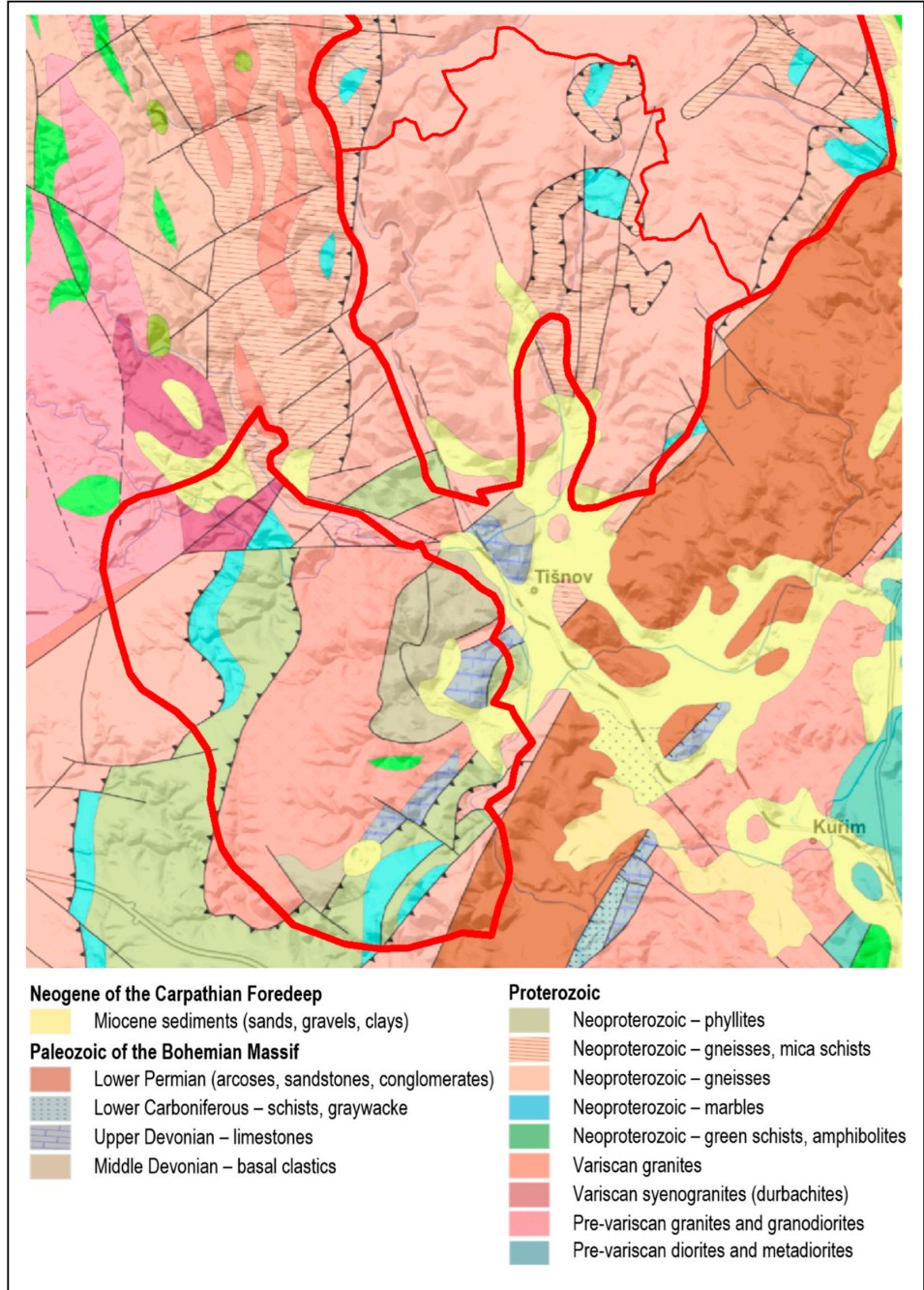

**Figure 2.** Geological settings of the Deblínská vrchovina Highland and the southern part of the Sýkořská hornatina Mountains (source data: [48,49]).

The detailed inventory and description of these areas were already undertaken [23,24,26,28]. The description of the specific sites which are important from the Earth-science point of view are included in the Database of Geological Localities [25], so only the brief characteristics of the study areas and selected sites of geotourist-interest are presented. The position of the sites can be seen in Figure 1.

2.2.1. Sýkořská Hornatina Mountains

The Sýkořská hornatina Mountains are situated 30 km north from Brno city, which is the second largest city in the Czech Republic (approximately 380,000 inhabitants, but the real number of people living here is higher). The harmonious landscape with well-conserved natural resources proves the

sustainable use of them and represents a good example of how the people exploited the landscape in the past (Figure 3a). The part of the area is legally protected within the Svratecká Hornatina Natural Park (which represents the lowest category of general nature conservation) and there are 12 sites protected within the category of National Reserve or National Monument [50]. Geologically, the area belongs to the northern part of the Svratka Dome [48]. The basement is rather monotonous and it is formed by biotite-muscovitic, sericite-muscovitic gneisses of the Bíteš group (Figure 3b) with limited occurrences of limestone and schists covered by quaternary sediments. In specific places, there are remnants of the marine sediments of the Ottnang age [49]. Despite the relatively monotonous geological composition, the morphological diversity of the area is very high. The landscape has been affected by several geomorphological processes, but the most significant landforms were created mainly by periglacial and cryogenic processes: Tors, castle-koppies, structural ridges, block accumulations and flows, nivation depressions, cryoplanation terraces, frost-riven cliffs, isolated boulders or congelifluction scree talus cones (Figure 3e). The anthropogenic landforms are present as well, especially those of agricultural origin (heaps, terraces, ramparts, and small walls). Due to the unique combination of geology and geomorphological landforms, the Sýkořská hornatina Mountains belong to the best-preserved areas with periglacial and cryogenic landforms in the Czech Republic [24].

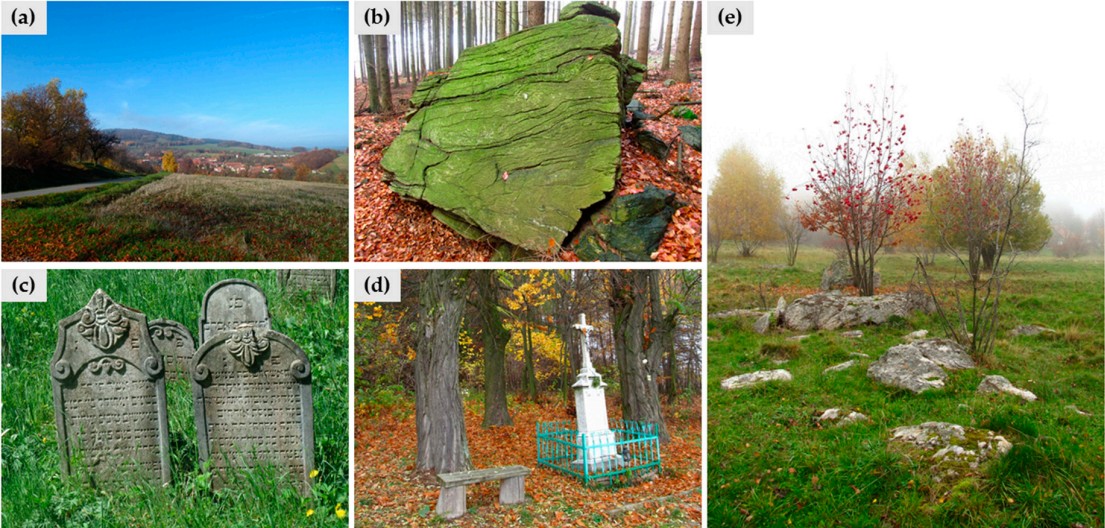

**Figure 3.** Sýkořská hornatina Mountains: (**a**) A view on the Synalov village (harmonic landscape with scattered settlements). (**b**) Bíteš orthogneiss—the main rock that builds up the area. (**c**) Jewish cemetery in Lomnice—an important part of cultural heritage with links to geodiversity (use of local stone for tombs); (**d**) small sacral monuments are common within the area and contribute to the typical character of the landscape; (**e**) congelifluction scree talus cones, isolated boulders and block accumulations on the slopes of the Sýkoř Hill (the highest peak of the study area).

The study area is rich in cultural features. The historically and architectonically valuable objects in the Lomnice Township on the southern part of the area (e.g., the Jewish cemetery (Figure 3c), synagogue, plaque column, castle, and church) and in the Lysice Township in the eastern part of the area (e.g., the chateau and church) are the most important. In the villages, sacral buildings, traditional agricultural buildings, and other objects of folk architecture can be found. In the open landscape, the small sacral objects, e.g., crosses or small chapels, are common (Figure 3d). Usually, the local building stone was used there [24].

Based on the literature review and fieldwork, the sites of geotourist interest were identified. Within the selected six sites (S1–S6, displayed in Figure 1), the above mentioned geological and geomorphological features and their relationships to the biodiversity and cultural heritage of the area can be observed.

Site 1 (S1)—Dobrá studně represents a complex of cryogenic landforms, especially the solifluction ones: Solifluction streams of several generations, terraces and occasional wet depressions can be found here. Locally, the massive gneiss boulders can be observed here. The largest solifluction stream is over 100 m long and 50 m wide, and together with others, indicates the existence of permafrost in the Pleistocene. The landforms have a crucial role for the differentiation of the vegetation cover; some of the endangered species live there.

As mentioned in the description of the geological settings of the area, the Bíteš gneiss is the main rock that builds the southern part of the Sýkořská hornatina Mountains. It has a porfyroblastic structure and it is clearly stratified with the clearly distinctive layers and direction of the metamorphosis [49]. The typical example of this rock can be observed on Hrušín (S2). The site is rich in cryogenic landforms (frost cliffs, boulder fields with plate boulders, debris accumulations, and cryoplanation terraces) and mezoforms of polygenetic origin: Small fissure caves, abris, mushroom rock, or bedding cavities. The block streams and debris accumulations are important from the ecological point of view: Thanks to the specific geomorphological and pedological settings, a natural debris forest with a high diversity of plants and the occurrence of protected species is conserved here.

The Sokolí skála Rock (S3) is a massive outcrop built of marginal facies of the Bíteš orthogneiss. The gneiss layers alternate with amphibolite beds there, which is important from the petrographic point of view. Besides this, the site possesses a significant geomorphological aspect: The outcrop was formed thanks to the erosional activity of the Svratka River which formed the deeply incised valley there. This valley is of epigenetic origin: During the tertiary uplift of the eastern margin of the Bohemian Massive, the Svratka River eroded the Miocene sediments, and then it continued to erode the gneiss bedrock. This is especially important from the paleogeographic point of view.

Synalovské kopaniny (S4) represents an example of congelifluction scree talus cones on the slopes of the Sýkoř Hill which are the result of Pleistocene cryogenic processes. Within the locality, the traces of recent slope movements can be observed. In the past, the site was used mainly as pasture land. Thanks to this, the typical mosaic of meadows, pastures, forests, and boulders has been conserved here until now.

Míchovec (S5) represents typical cryogenic landforms of the area: Tors, nivation depressions, and block streams. The cryogenic landforms are similar to those in other localities, but thanks to specific microclimatic conditions, the nivation processes were relatively intensive and strong here—there are several nivation depressions with abri with a height over 4 m. Besides this, numerous fissure caves can be found there, and recumbent folds are observable on the walls of frost cliffs. The site is also important from the ecological point of view: The occurrence of well-conserved debris forests with a massive population of endangered species *Lunaria rediviva*.

Veselský chlum (S6) displays specific aspects of the study area's history and shows evidence of how the people in the past used the land and natural resources. Numerous anthropogenic landforms (especially agrarian terraces, ramparts, and unpaved walls made of flat gneiss stones collected from the surrounding fields and pastures) can be found here. The site is protected by law and the reason for protection is the well-conserved segment of the harmonious cultural landscape with a unique mosaic of pasture land, orchards, scattered greenery, and anthropogenic landforms with high aesthetic value. Moreover, the site is an important viewpoint geosite (as defined by [51]): It offers a view on the Svratka River valley and its surroundings, so the geomorphological context of the study area can be studied and observed here.

### 2.2.2. Deblínská Vrchovina Highland

Deblínská vrchovina Highland lies about 25 km northwest from the Brno city. The area has a very varied geology, thanks to its position on the eastern margin of the Bohemian Massif. High lithological diversity implies a high diversity of landforms and processes. The area represents the harmonic landscape characterized by a mosaic of fields, forests, meadows, and ancient orchards. The southern margin of the area is a part of Bílý potok Natural Park [50]. The only Nature Reserve situated in the study area is represented by beech forests at Slunná; however, numerous geological and geomorphological

sites (rock outcrops and abandoned quarries) are included in the Database of Geological Localities [25]. Currently, the area represents similar recreational and touristic background for the Brno City as Sýkořská hornatina Mountains (described in Section 2.2.1 Sýkořská Hornatina Mountains); however, they both remain in the shade of popular, and geologically and geomorphologically spectacular Moravian Karst [52] which is visited more frequently.

The area is situated in the southern part of the Svratka Dome, a structure which includes Svratka massif (composed of the oldest rocks of the area: Prepalaeozoic intrusive and metamorphic rocks, Devonian basal clastics and limestone, and Carboniferous siliciclastic), and the Moravicum nappe which is made up of a weak metamorphosed volcano-sedimentary complex with prevailing phyllites and orthogneiss (metamorphosed Cadomian granite) [48,49]. Neogene is represented by Miocene and Pliocene freshwater sediments that fill older valleys and depressions between the Maršov and Lažánky villages. Here, the lower Miocene sediments with abundant fauna are overburden with clays, sands, and gravels [53]. Pleistocene is represented by fluvial sandy gravel, which often forms terraces at different heights above the present valley bottom. Loess sediments are also common and reach the thicknesses of up to 5 m. Holocene flood sediments are not very thick (maximum 2 m). The Holocene also includes anthropogenic sediments (heaps and dumps of the quarries) [53].

The fluvial, karst and anthropogenic landforms, together with polygenetic rock formations modeled by slope and cryogenic processes, represent the most significant features of the study area [23,26]. The origin of the remarkable landforms is often linked to the lithology; e.g., the resistant rocks (limestone, basal clastics, quartzite, and gneiss) formed significant outcrops and elevations (Figure 4a). The most important fluvial landforms are represented by the Svratka Valley (Figure 4b); typical fluvial mezoforms can be observed in Svratka's tributaries. Anthropogenic landforms are represented by the abandoned kaolin pit (Figure 4c), limestone quarries (Figure 4d), and remains from the medieval mining of ores (adits and heaps). The use of limestone can be traced back to the Middle Ages and until the present; the remains of old lime kilns are preserved (Figure 4e) and represent an important part of local cultural heritage [23]. Water management landforms are related to the streams and allow tracing the use of natural resources in the past (Figure 4f). Other cultural features of the area are represented by historical buildings in the Tišnov city (situated on the border of the area) and Předklášteří village (especially Cistercian convent Porta Coeli) where the local building stone and material from nearby quarries were used.

Based on the literature review and fieldwork, the sites of geotourist-interest were identified. The selected sites (D1–D7, displayed in Figure 1) allow observing and studying specific geological and geomorphological features, and their relationships to the cultural heritage and ecological aspects.

Site 1 (D1)— the rock outcrop Skalky, is built of resistant quartzite. Geomorphologically, it can be described as a monadnock. Similar outcrops are situated approximately 700 m southwest of the site, in the valley of Salašský potok Stream. There, they form natural steps and during the wet seasons, there are small waterfalls. The position of this lithological member of the parautochtonal Svratka Dome sediments is not clear yet, so the site is important as a study locality. Generally, these outcrops represent a typical example of selective erosion and on the surface; numerous meso- and microrelief phenomena (especially small caverns filled with calcite and baryte) can be seen.

The kaolin deposit in the old kaolin pit (D2) is situated on the contact zone of granodiorite and phyllites. Kaolin was exploited here at the beginning of the 20th century, but it was stopped in 1939 because of the bad quality of the material. During the active exploitation, several prospection shafts were dug there, and several lignite seams were discovered. Adjacent sediments (clays) are paleontologically rich and accompanied by gravels. They represent a relic of ancient valley fill. The site is interesting from the geomorphological point of view: Small abrasion cliffs and landslides can be observed on the pit slopes.

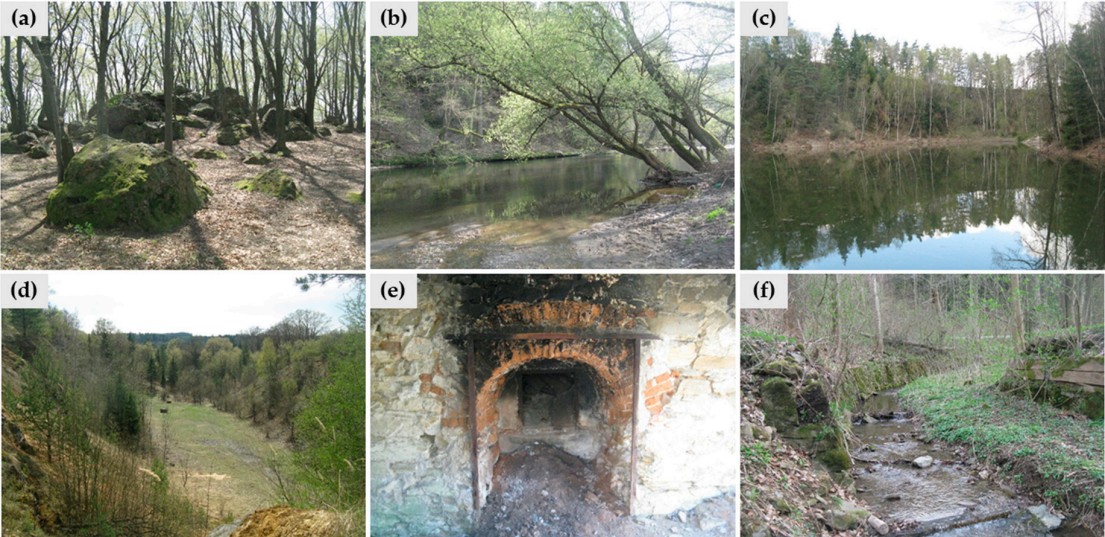

**Figure 4.** Deblínská vrchovina Highland: (**a**) Quartzite outcrops of Skalky—illustration of the role of the resistance of the rock. (**b**) Svratka River valley with gravel banks. (**c**) The flooded kaolin pit near Maršov; (**d**) an abandoned limestone quarry—one of the sites where the limestone for lime burning was extracted; numerous karst features are present there. (**e**) Remains of the Havílkova lime kiln near Lažánky, an important part of industrial heritage; (**f**) water management anthropogenic landforms—channels and water races were used by mills.

The evidence of limestone quarrying is represented by abandoned limestone quarry (D3). Within the area, there are several old quarries that are currently very well incorporated into the landscape, and increase landscape diversity. Slight karstification can be observed here, including karren, small caves, and cavities filled with calcite. The limestone extracted here was suitable for lime burning; near the site, an old lime kiln is situated. The site is thus important from the historical point of view as it brings forth the evidence of using the natural resources in the past.

Under the active limestone quarry (currently closed for public), on the right slope of Maršovský potok Stream, the karst spring is situated (site D4). It is probably connected with cave systems situated in the active quarry because during dry periods, the cavemen found the continuation towards the active quarry. In this quarry, (situated just several tens of meters north of the spring), several caves with were found and documented in the 1980s, but due to the progressive quarrying, these caves were destroyed. However, the existence of the spring, its hydrological aspects, and its continuation into the limestone massive suggest that there are uncovered cave systems situated beneath the current level of the lowest quarry bench.

The site D5 (Maršovský potok and Pejškovský potok streams) represents the complex of fluvial landforms. Both valleys are rich in meanders, empty oxbow lakes, cutoffs, alluvial ramparts, gravel banks, and other fluvial landforms. At Maršovský potok Stream, there is an observable alteration of floodplains and deeply incised segments of the valley, which follow the alteration of bedrock. Moreover, the traces of anthropogenic use of the watercourses can be seen here (old water races and small dams)

The Svratka Valley (D6) is an epigenetic valley where the relics of fluvial terraces in different heights above the present valley bottom can be found. Thus, the site has high paleogeographic importance. The site is also interesting from the geomorphological point of view: Numerous cryogenic landforms (frost cliffs, and boulder and debris accumulations) are situated here. Moreover, specific vegetation communities with the occurrence of rare and endangered species can be found here. Several thermophilic species have the northernmost border of their areal here thanks to the specific geomorphologic and climatic conditions (dry and steep southwestern slopes without forest).

The Vokoun's viewpoint (D7) represents a viewpoint geosite which allows for the observing of the Tišnovská kotlina Basin (with the Svratka floodplain, Květnice Hill and Dřínová quarry which are important from an Earth-science point of view, but situated outside the study areas) and a southern part of the Sýkořská hornatina Mountains. The viewpoint is situated on the steep slope on the southern end of the village of Předklášteří, not far from an old gneiss quarry. The terrain was badly accessible and the view was obstructed by trees; however, thanks to the activity of local enthusiasts, the tourist facilities (steps, shelter, and information panel) were constructed and a newly marked tourist path leads there.

Both areas (Sýkořská hornatina Mountains and Deblínská vrchovina Highland) were recently the subject of several large-scale paintings of Adam Kašpar who introduced them at a temporary exhibition in Tišnov. They have also been the subject of other painters in the past (e.g., J. Jambor). An important social event related to geodiversity is represented by traditional mineral exhibitions which are held two times per year in Tišnov.

## 3. Results

The study areas and the sites of geotourist-interest were assessed by using the methods described in chapter 2.1. The assessment of the sites of geotourist interest is presented in Table 3; the assessments of the areas were elaborated separately (Table 4 for Sýkořská Hornatina Mountains and Table 5 for Deblínská vrchovina Highland). The SWOT analysis was elaborated for the two territories together, as they are situated close to each other and most of the characteristics of Sýkořská hornatina Mountains and Deblínská vrchovina Highland are in common or very similar. Table 6 thus presents the basic SWOT analysis for both areas; Table 7 shows the extended SWOT analysis. Where the differences between the particular areas occur, they are marked by indexes (S for Sýkořská hornatina Mountains, and D for Deblínská vrchovina Highland).

**Table 3.** Assessment of the sites of geotourism interest in the Sýkořská hornatina Mountains (S1—S6) and Deblínská vrchovina Highland (D1—D7).

| Criterion/Site | S1 | S2 | S3 | S4 | S5 | S6 | D1 | D2 | D3 | D4 | D5 | D6 | D7 |
|---|---|---|---|---|---|---|---|---|---|---|---|---|---|
| integrity | 1.00 | 1.00 | 1.00 | 0.75 | 1.00 | 0.75 | 1.00 | 0.50 | 0.50 | 0.75 | 0.75 | 1.00 | 1.00 |
| representativeness | 1.00 | 1.00 | 1.00 | 0.75 | 1.00 | 1.00 | 1.00 | 1.00 | 1.00 | 1.00 | 1.00 | 1.00 | 1.00 |
| rareness | 0.75 | 0.25 | 0.75 | 0.50 | 0.50 | 0.50 | 0.75 | 1.00 | 0.25 | 0.50 | 0.25 | 0.75 | 0.50 |
| paleogeographical interest | 0.75 | 0.50 | 1.00 | 0.50 | 0.75 | 0.50 | 1.00 | 0.75 | 0.50 | 1.00 | 0.50 | 1.00 | 0.50 |
| **scientific value (synthesis)** | **0.88** | **0.69** | **0.94** | **0.63** | **0.81** | **0.69** | **0.94** | **0.81** | **0.56** | **0.81** | **0.63** | **0.94** | **0.75** |
| ecological | 1.00 | 1.00 | 0.50 | 1.00 | 1.00 | 1.00 | 0.25 | 0.50 | 0.50 | 0.50 | 0.50 | 1.00 | 0.50 |
| aesthetical | 0.50 | 0.50 | 0.75 | 0.75 | 0.50 | 1.00 | 0.50 | 0.75 | 0.50 | 0.25 | 0.50 | 0.75 | 1.00 |
| cultural | 0.00 | 0.00 | 0.00 | 0.75 | 0.00 | 1.00 | 0.00 | 1.00 | 1.00 | 0.25 | 1.00 | 0.50 | 1.00 |
| **added value (synthesis)** | **0.50** | **0.50** | **0.42** | **0.83** | **0.50** | **1.00** | **0.25** | **0.75** | **0.67** | **0.33** | **0.67** | **0.75** | **0.83** |
| protection status | 1.00 | 1.00 | 1.00 | 1.00 | 1.00 | 1.00 | 0.50 | 0.00 | 0.00 | 0.00 | 0.00 | 0.00 | 0.00 |
| threats | 0.50 | 0.75 | 0.75 | 0.50 | 0.75 | 0.50 | 0.75 | 0.25 | 0.50 | 0.50 | 0.50 | 0.25 | 0.50 |
| accessibility | 0.50 | 0.50 | 0.25 | 0.75 | 0.50 | 0.75 | 0.75 | 0.50 | 0.50 | 0.50 | 1.00 | 0.50 | 0.75 |
| security | 0.75 | 0.75 | 0.50 | 0.75 | 0.75 | 1.00 | 1.00 | 0.50 | 0.50 | 0.25 | 1.00 | 0.25 | 0.75 |
| site context | 0.50 | 0.50 | 0.75 | 0.75 | 0.50 | 1.00 | 0.75 | 0.75 | 0.75 | 0.25 | 0.50 | 0.50 | 1.00 |
| tourist infrastructure | 0.25 | 0.25 | 0.25 | 0.50 | 0.25 | 0.50 | 0.75 | 0.25 | 0.25 | 0.50 | 0.75 | 0.75 | 1.00 |
| interpretive facilities | 0.25 | 0.50 | 0.25 | 0.50 | 0.50 | 0.50 | 0.00 | 0.00 | 0.25 | 0.00 | 0.50 | 0.25 | 0.75 |
| educational interest | 0.50 | 0.75 | 0.50 | 0.50 | 0.50 | 0.50 | 0.50 | 0.75 | 0.75 | 0.50 | 0.75 | 0.75 | 0.75 |
| **use characteristics (synthesis)** | **0.53** | **0.63** | **0.53** | **0.66** | **0.59** | **0.72** | **0.63** | **0.38** | **0.44** | **0.31** | **0.63** | **0.41** | **0.69** |

**Table 4.** Assessment of the Sýkořská hornatina Mountains.

| | Criterion | Qualitative Assessment |
|---|---|---|
| scientific value | integrity | The current status of the study area is good and it represents a typical example of the sustainable and regardful use of the natural resources (both in past and present), the current status of Earth-science phenomena in general is good as well. |
| | representativeness | The area represents a well-conserved landscape with traces of past sustainable use of natural resources. Particular sites represent typical examples of cryogenic, fluvial and anthropogenic landforms and processes. |
| | rareness | Similar harmonic landscapes can be found in different areas within Moravia, so the degree of rarity is not high. However, the geomorphological diversity is high as numerous landforms are concentrated at relatively small area. |
| | paleogeographical interest | n/a |
| added value | ecological | Most of the landscape segments which are legally protected are home to the specific and rare species, so the ecological value of the study area is quite high. |
| | aesthetical | Within the study area, there are numerous viewpoints to the open landscape. The landscape pattern is quite diverse (small pieces of fields, forests, villages, meadows, alleys), so the study area is aesthetically attractive. Moreover, there are not any large constructions which would disturb the landscape character. |
| | cultural | Cultural features are concentrated in the settlements, there are numerous small sacral objects both in the villages (chapels) and in the open landscape (wayside crosses). Also, there are old agricultural buildings and other objects of folk architecture. The landforms of anthropogenic origin (especially agricultural landforms) are important from the historical point of view as they serve evidence of the use of the landscape in the past. A series of paintings of young artist Adam Kašpar and some paintings of Josef Jambor (Moravian landscape painter) reflect the geodiversity of the area. Existence of several legends about Sýkoř Hill. |
| use characteristics | protection status | The conservation of the specific geological and geomorphological phenomena and adjacent ecosystems is adequate—most of these landforms are protected within the category of Natural Reserve or Natural Monument. |
| | threats | Concerning the environmental, respectively geological and geomorphological hazards, the area is not at risk. There may be anthropogenic pressure connected to the construction activity (new houses, communications). |
| | accessibility | The public transport is sufficient as the area is partly included in the Integrated transport system of the South-Moravian region. The permeability of the landscape is quite good thanks to the presence of the network of paths and local communications (both marked and not marked). |
| | security | n/a |
| | site context | n/a |
| | tourist infrastructure | Some of the marked paths lead through the most attractive segments of the area accompanied by shelters. The limited accommodation capacities in Lomnice or Lysice. As the area is rather used for one-day trips, the current tourist infrastructure is relatively sufficient. There are local restaurants even in the smaller villages. |
| | interpretive facilities | The area is promoted especially via web pages of the local communities and web pages devoted to the touristic attractions of the South-Moravian region. The knowledge and popularity of the area are rather local/regional (it is not well-known on the national level). |
| | educational interest | The cryogenic landforms are well visible (especially during the season without vegetation) and if the short explanation is given (e.g., via information panels), they are also comprehensive for the public. Anthropogenic landforms and processes are also easy to understand as they are related to the common activities of humans (e.g., picking the stones from the fields and accumulating them on the agrarian heaps or ramparts). |

**Table 5.** Assessment of the Deblínská vrchovina Highland.

| | Criterion | Qualitative Assessment |
|---|---|---|
| **scientific value** | integrity | The current status of the landscape is relatively good, however, particular sites can suffer from human activities (e.g., active quarrying, transport, agriculture, expansion of buildings into the open landscape). |
| | representativeness | The area represents a relatively well-conserved landscape with traces of past sustainable use of natural resources (limestone, kaolin, water resources). Thanks to the high lithological diversity, the morphological diversity is also high (at a relatively small area, there are landforms of different origin). |
| | rareness | Similar type of landscape can be found in different areas within Moravia, so the degree of rarity is not high. However, at specific sites, the rareness of the Earth-science phenomena can be considered high at regional level. |
| | paleogeographical interest | n/a |
| **added value** | ecological | Numerous geosites are accompanied by important ecosystems and protected species. The abandoned quarries also play a specific role regarding the biodiversity and ecosystems. Karst caves and old mining landforms (adits) are home to the protected species (bats). |
| | aesthetical | The mosaic of fields, meadows, and forests is aesthetically valuable, abandoned quarries increase the overall diversity of the landscape. The landscape character is disturbed by extensive built-up areas (inadequate development of living, land occupation) and partly by quarrying. |
| | cultural | There is a lot of buildings that use local stone. Generally, the cultural heritage is concentrated in the Tišnov city and Předklášteří village (convent Porta Coeli) and small sacral buildings in the villages within the study area. Thanks to the historical exploitation of natural resources (limestone, ores) and partly conserved mining landforms, the cultural value is also very high. A series of paintings of young artist Adam Kašpar reflect the geodiversity of the area. |
| **use characteristics** | protection status | The southern part of the area is protected in the category "Natural park"—the category of general protection of nature. The protection of the geological and geomorphological phenomena is not sufficient (particular sites are included in the Database of CGS, but they have no legal protection with the exception of karst caves which are generally protected by law). |
| | threats | Abandoned quarries are often used as dumps and suffer from vandalism. These undesirable activities can affect or damage natural karst features. Spreading the area of the fields can disturb the harmonic landscape as well as the spreading of the family houses and intensifying the transport. |
| | accessibility | The public transport is sufficient as the area is included in the Integrated transport system of the South-Moravian region. The accessibility to the particular sites is in most cases easy, the terrain is not difficult. The permeability of the landscape is quite good thanks to the presence of the network of paths and local communications (both marked and not marked). |
| | security | n/a |
| | site context | n/a |
| | tourist infrastructure | Tourist paths lead through the area, however, some geosites remain out of the reach of these paths. Accommodation and catering are accessible especially in Tišnov and Veverská Bítýška, but in small villages too. The area is usually visited within one-day trips, so currently, the tourist infrastructure is sufficient. |
| | interpretive facilities | The area is promoted especially via web pages of the local communities and web pages devoted to the touristic attractions of the South-Moravian region. The knowledge and popularity of the area are rather local/regional (it is not well-known on the national level). Specific sites are well promoted on local guides and websites of the municipalities, but some sites with high scientific and added values remain "unexplored." |
| | educational interest | Karst, fluvial and other features that are present here, are not important in size, but they can provide a solid basis for explanation and educational activities for local schools. Abandoned quarries are a good example of using natural resources in the past and together with cultural aspects can be an important resource for education. |

**Table 6.** The basic SWOT analysis for both study areas.

| Strengths | Weaknesses |
|---|---|
| 1. harmonic landscape with well-conserved nature, high lithological (D) and morphological diversity | 1. the tourist infrastructure is not sufficient if the visitors want to spend here more time |
| 2. landforms and processes are well visible and comprehensible for the public | 2. the educational, recreational and tourist potential is not still fully recognized by locals |
| 3. high added values (cultural, ecological) | 3. the geodiversity is not promoted as a resource for tourism and education |
| 4. marked paths leading to the most attractive natural features, good permeability of the landscape | 4. landscape and landforms have been affected by anthropogenic activity, e.g., active quarrying, urban spreading, transport or agriculture (D) |
| 5. the areas do not suffer from excessive attendance | 5. lack of legal protection of the geological and geomorphological phenomena (D) |
| 6. good accessibility by public transport | 6. lack of interest from the municipalities for the geotourism-development (the geotourism-development is not the priority of the local stakeholders) |
| 7. sufficient legal conservation of specific sites (S) | |

| Opportunities | Threats |
|---|---|
| a. study areas as a good option for one-day trips from the Brno city | a. the fast and inadequate development of the tourist infrastructure can cause the disturbances and damages to the landscape and particular geological and geomorphological phenomena |
| b. both areas can be seen as an alternative to overcrowded Moravian Karst situated nearby | b. the continuing anthropogenic activity (inadequate land use) can negatively affect the character of villages or generally, the harmonic character of the landscape and it can change the aesthetic quality of the landscape |
| c. promotion of close links between geodiversity and culture/history can raise the awareness of geodiversity and foster the local identity | c. further preference of construction activity before nature conservation and sustainable development |
| d. geotourist and geoeducational potential can be used both for the lay public (visitors) and organized groups of students of local/regional schools | |
| e. reasonable developing of the geotourism as a driving force for the local economic development | |
| f. possibility to cooperate with local communities, schools, voluntary associations of the municipalities or subjects within Local Action Groups etc. | |

**Table 7.** The extended SWOT analysis for both study areas.

| S-O Strategy (maxi-maxi) | W-O Strategy (mini-maxi) |
|---|---|
| - promotion of the natural and cultural heritage related to geodiversity can attract visitors from nearby towns and metropolis (1, 2, 3, a, b) | - focus on the short-term recreation and tourism (1, a, b) |
| - both areas can be presented as an alternative to overcrowded Moravian Karst (5, b) | - promotion of the links between abiotic-biotic-cultural components of the area can help to raise the awareness of the geotourist and geoeducational resources (2, 3, 6, c, d) |
| - good accessibility and good permeability of the landscape can be presented as an advantage and interesting investment opportunity for the potential "developers" of tourist infrastructure (4, 6, e) | - promotion of the geotourist resources and their good accessibility as an advantage for tourism and economic development (3, 4, e, f) |
| - developing a new geotourist product or including the geodiversity aspects into the current tourist offer can be used both by visitors/tourists and by students and pupils of local schools and schools (1, 2, d, f) | - involving local communities and subjects (e.g., schools) to raise the awareness of the value of geodiversity (3, 4, d, f) |
| | - emphasizing the close links between abiotic—biotic—cultural components of the landscape can help to justify the need for conservation (4, 5, c) |

| S-T Strategy (maxi-mini) | W-T Strategy (mini-mini) |
|---|---|
| - present the geodiversity as an important resource for tourism and as an entity that has to be conserved for future generations (1, 2, a, b) | - to avoid the anthropogenic pressure and uncontrolled development of tourist infrastructure, especially via landscape planning, development strategies and conservation measures (1, 4, a, b) |
| - maintaining and fostering sufficient legal conservation of the specific sites can help to avoid the disturbing activities that can negatively affect the landscape and particular geodiversity components (7, b) | - to promote geotourism concept as an alternative to traditional (demanding) tourism, to stress the sustainability of this form of tourism, |
| - promotion of geotourist resources (incl. examples of good practice from different regions) on the meetings of municipalities or local stakeholders (e.g., Local Action Groups) can overcome the lack of interest from the local stakeholders (1, 2, c) | - to cooperate with successful subjects and regions that use the concept of geotourism (2, 3, 6, c) |
| | - maintain the development of tourist infrastructure in accordance with geotourism and nature conservation principles (4, 5, a, b) |
| | - to avoid the future damage of geological and geomorphological sites by using the landscape planning and management strategies and public discussion with stakeholders (4, 5, 6, b, c) |

## 4. Discussion and Concluding Remarks

The criteria used within geomorphosite concept proved to be a simple and comprehensive tool for qualitative and semi-quantitative assessment of geotourist resources within larger areas. The assessment of larger (wider) areas within a geomorphosite concept has some specifics—this assessment is rather qualitative and was based on expert knowledge, numerical assessment of particular sites, detailed fieldwork, or discussions with residents. A degree of subjectivity exists there; however, the qualitative assessment is probably more comprehensible for the local authorities or stakeholders than the numerical one. In the future, the assessment of geotourist resources can be accompanied, e.g., by an approach presented by Martins and Pereira [33], which is based on the perception of local people. The numerical assessment of the sites of geotourist interest which served as one of the bases for the qualitative assessment are more objective; however, the assessment of specific criteria within the Reynard's method remains relatively subjective (e.g., aesthetic value).

In comparison with the geomorphosite concept, the SWOT analysis represents an even more comprehensible tool for assessing geotourist resources. It is easily understandable for authorities, members of Local Action Groups, and other subjects that aim to participate in geotourism development, and thus can serve a simple way for assessing geotourist resources and setting the directions and possibilities of geotourism development as proved by numerous studies; e.g., [21,44,46,54,55].

Qualitative and semi-quantitative assessment, basic SWOT and extended SWOT analysis thus allowed us to identify the directions of geotourism development, and to propose particular activities to use the geotourist resources in a sustainable way. Based on this, specific strategies for geotourism development can be proposed:

1) The geodiversity of the assessed areas is not unique on the national level, but the educational value is high: Landforms and processes are illustrative, visible, and relatively simple to understand (e.g., thq role of the rock resistance in the shaping of significant outcrops at Skalky in the Deblínská vrchovina Highland, or typical cryogenic landforms in the Sýkořská vrchovina Mountains) which is supported by numerical assessment of specific localities. Integrity and conditions of landforms are relatively high thanks to the position of the areas outside the main tourist destinations. The landscape is well-preserved and it shows a good example of the co-existing of man and nature. Moreover, specific sites are very important from the paleogeographic point of view (especially epigenetic valleys of Svratka). These issues were assessed as the main resource with high potential for developing sustainable geotourism and educational activities (both for local people and visitors). It has to be emphasized that geotourism provides economic, cultural, and social benefits for both visitors and hosting communities [56].

2) Added value is closely linked to the geodiversity. Both areas can present numerous examples of the mutual relationships between abiotic, biotic, and cultural components of the landscape (historical values, geomythological aspects, the traces of the landscape memory, and local materials used for local buildings and constructions). This is supported by the high value of several sites—especially in the Sýkořská hornatina Mountains, where numerous sites have an important biotic element which is legally protected there (together with geo-elements). This holistic approach has to be taken into account when planning the management of the landscape and conservation measures: The existing links can help improve acceptance of conservation measures (in Deblínská vrchovina Highland) and can increase the overall attractiveness of the area in the terms of interpretation of the heritage. The mutual links between abiotic, biotic, and cultural components can be used for environmental education as well, and can help to raise awareness about geodiversity in the study areas.

3) Accessibility of the areas is relatively good; the tourist facilities are average or below the average. It is subject to further discussions about whether the adjacent tourist infrastructure has to be developed. If decided to support the geotourism in these areas, some additional tourist infrastructure should be built; however, this has to be balanced with geoconservation

principles. According to Dowling and Newsome [18], the geotourism should be sustainable and environmentally friendly, so this has to be respected while developing the tourist infrastructure, improving the access to the particular sites or building accommodation capacities.

4) The number of visitors and knowledge/popularity of the areas is not high. The promotion is very irregular. In order to develop the geotourism, the promotion should be assured and should take into account two aspects: The promotion of specific sites of geotourist interest and the promotion of the area as a whole with its cultural heritage related to geodiversity, with its history or with its specific characteristics. Due to the fact that the geotourist resources of these areas cannot compete with the Moravian Karst Protected Landscape Area with its caves, springs, and spectacular outcrops and valleys, these areas will probably never reach high popularity. Nevertheless, they can be promoted as a calm alternative to the overcrowded Moravian Karst or an accessible and pleasant area for short-term recreation and tourism. New geotourism products (e.g., educational path connecting significant sites of geological and geomorphological interest, local products related to the geodiversity resources, and information panels on websites) can attract both visitors and local people.

5) If it is decided to support the development of geotourist activities, close communication with local communities and initiatives is needed in order to develop the effective management of geotourist resources. Cooperation with research institutions is important, as academic research provides the background for further activities supporting the promotion of geoheritage [57]; however, they have to be implemented by local communities themselves. Thus, a bottom-up approach has to be respected. Moreover, the volunteer activities can increase the local awareness and appreciation of geoheritage [58], and can foster the local identity in general. As there are active NGOs, volunteer associations of municipalities, or Local Action Groups in these areas (as indicated in SWOT analysis), it can be supposed that the bottom-up approach can have success. There have already been several specific cases recorded where the local NGOs made the sites accessible or visible.

6) The geotourist activities have to accept the intrinsic value of geodiversity and respect the principles of nature conservation (respectively geoconservation as defined by Prosser et al. [59]). Legal protection of specific sites is already set up (in Sýkořská vrchovina Mountains); however, other sites are not protected, thus can be endangered by human activities. The involvement of local subjects, and informing them about these geotourist resources can improve acceptance of the conservation measures. As the geoethical practice is an essential part of geotourism [56], this aspect of using the geotourist resources should be also taken into account.

As geodiversity represents a basis for the geotourism, it can be considered an important resource for the local and regional development. In order to use this resource in a responsible and sustainable way, the inventory and assessment of the geodiversity and geoheritage are the initial steps which have to be reflected in the plans for geotourism-development; e.g., [18,60,61]. In these terms, cooperation with universities and research institutions is more than desirable.

The particular outcomes from the assessment and basic/extended SWOT analysis can be used in strategic development document or planning. The plans have to follow the geoconservation rules and principles of sustainable development, and in the future, they can become a respected part of a local and regional planning and development conceptions and strategies.

**Funding:** The research was supported by Internal Grant Agency MENDELU, project number VP_2018024.

**Acknowledgments:** The Author thanks the three anonymous reviewers for their helpful comments.

**Conflicts of Interest:** The author declares no conflict of interest.

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
