# Peer review of "Assessing Geotourism Resources on a Local Level: A Case Study from Southern Moravia (Czech Republic)"

_resources, doi:10.3390/resources8030150_

Round 1

Reviewer 1 Report

The article focuses a very interesting topic. It can be useful for the territory valorisation, especially in depressed economic areas. My main concern is related to the methodology adopted by the author. It is a qualitative approach, naturally subjective. Some works whose theme is similar have adopted other type of methodologies, such as the perception of local inhabitants, for example. In this sense, I believe that the work would gain strength if another methodological approach were adopted.

In the introduction section it would be important to consider some more recent work like Ólafsdóttir, R. (2019) doi:10.3390/geosciences9010048; Martins, B. & Pereira, A. (2018) DOI: 10.3390/geosciences8100381 or Adriana-Bianca Ovreiu, Adriana-Bianca et al (2019) DOI: 10.1007/s12371-019-00352-7. 

They give a holistic and integrated approach to geotourism.

Figure 2 should be edited. It must include the source. It would also be convenient to introduce chronostratigraphic series of lithology in the legend.

Author Response

Dear Reviewer,

thank you for the comments, please see the attachment.

Reviewer 2 Report

Overall, this is quite clearly and well written paper which deserves publication after some minor-to-moderate revision.

My two main points of concern are: (a) entirely qualitative nature of the paper and hence, tables 3 and 4 provide limited insights into the specific characteristics of the areas and diversity within these areas; (b) specific localities are hardly referred to, even those of significant value. For both reasons, the paper remains at rather general, close to superficial level and information contained may be difficult to be used in any comparative analysis in the future (with other areas). It would be good if some kind of semi-quantitative assessment is undertaken, to support information in Tables 3 and 4 (which, importantly, at present largely repeat the text and offer little new information), and certainly point (b) can be addressed by naming/briefly characterizing key localities. They can be listed in a new table and should be indicated on Fig. 1.

Specific comments:

L77-80 – this is unclear and requires rewriting. I cannot follow the logic here. Why threats to individual sites justifies assessment at the landscape scale (wider areas)?

L109-110 – but these papers are hardly accessible, so more information here would be helpful to an interested reader

L159 – “implies” (not “implied”)

L166 – provide references to geotourism use of Moravian Karst and its geodiversity

L178-179 – rephrasing recommended, to avoid “landforms” four times in one sentence

L182ff – photographs are incorrectly referred to, in wrong order (we have ‘a’ first, followed by ‘f’, then ‘e,f’ etc.). Since all photos are of identical dimensions, the easiest way is to rearrange the collection of photographs.

L248 – “and” (not “an”)

L280 – I am not sure why just these papers were selected for publication. Are they specifically recommended? I think we do not need references in this part of the paper.

Table 3 – description of Earth science values seems contradict L136-137 (high value of cryogenic landforms at the national level)

Here I also wonder if some further comments on visibility can be offered. I happened to visit the area some years ago, in early April, and at that time cryogenic landforms were well seen as foliage was minimal. But it may not be so in summer, with peak of vegetation. This is a factor to consider. I am not aware about too many papers addressing this issue but something can be found in: Visitors' background as a factor in geosite evaluation : the case of Cenozoic volcanic sites in the Pogórze Kaczawskie region, SW Poland. Geoturystyka (Geotourism), 38–39: 3–18.

Table 4 – „a lot of sites that are important” is mentioned under Earth science values – this links with my general comment at the beginning (point b). This is too general and of little value for the reader. Please name these important localities in text or preferably another table and cross-refer to that table here.

Table 5

-          W4 sounds like threat, not a weakness

-          Od – “lay people” (not “laic”)

-          Oe – I am afraid this is applicable to any locality, I would delete it

-          Tc – I think that “lack of interest” is a weakness and this may lead to threat which is “Preference of construction activity…” (cause and reason)

And we have a few contentious issues at the end of the paper and the author may like to consider the following suggestions:

L253 – this is debatable that focus should not be on individual localities. Still, people go to see localities, especially in geotourism, and these are the fundamental resources. In my view, a selection of key geosites from both areas should be selected and promoted, if geotourism development is a goal.

L255 – another debatable claim. Although Moravian Karst will probably indeed remain always a more popular area, it does so because it has a clear identifier (brand name) which is “Karst”. The areas reported in the paper do not have such, but they may have, perhaps especially the harmonious cultural landscape of Sykorska hornatina (focus on cryogenic inheritance and cultural landscapes). It is a matter of finding the catchy brand name. The author may wish to have a look at this recent publication: Promoting and interpreting geoheritage at the local level – bottom-up approach in the Land of Extinct Volcanoes, in Geoheritage 2019 (available online) (and further references there) for one comparable example, but I am sure that more can be found.

L261 – this is very general statement, applicable to everywhere. Are there any signs of interest among the locals to promote geotourism? Does bottom-up approach have a chance to succeed?

Fig. 2 – indicate the extent of two study areas, as shown on Fig. 1 and add selected place names.

Fig. 3 – rearrange the order of photographs, to achieve consistency with text.

Author Response

Dear Reviewer,

thank you for the comments, please see the attachement.

Reviewer 3 Report

This paper is an application a known method (SWOT analysis) to territories of Czech Republic. It is well presented even if I think can be improved after reorganization of chapters and more,  inserting a proper Discussion chapter.

Suggestions:

1.       Introduction

The introduction is quite well organised. My concern is that there are only few words on the aims of this paper: this part should be improved.

2. Materials and Methods

- Field work and GIS analysis were developed in previous works, therefore this paper looks as a deeper/larger analysis of something already known in literature.

2.2 -> 3. Study Area

It is better to separate this chapter from the previous making a new one.

3 -> 4

This chapter should re-titled “SWAT analysis” (instead of Results) because is the application of this matrix to this real case

5. DISCUSSION

This chapter is a missing: the paper will benefit of a broader discussion instead of a fast resume in the Concluding remarks.

Round 2

Reviewer 1 Report

Although the methodology adopted by the author has some limitations, the current version of the work has improved substantially.
In this sense the article presents conditions to be published. Figure 1 can be improved.